# Prevalence and Subtype Distribution of *Blastocystis* sp. in Diarrheic Pigs in Southern China

**DOI:** 10.3390/pathogens10091189

**Published:** 2021-09-14

**Authors:** Pei Wang, Sen Li, Yang Zou, Zhao-Wei Hong, Ping Wang, Xing-Quan Zhu, De-Ping Song, Xiao-Qing Chen

**Affiliations:** 1Jiangxi Provincial Key Laboratory for Animal Health, College of Animal Science and Technology, Jiangxi Agricultural University, Nanchang 330045, China; 15779520069@163.com (P.W.); lisentdcq@163.com (S.L.); H554526@163.com (Z.-W.H.); jxjs6263wplm@163.com (P.W.); 2State Key Laboratory of Veterinary Etiological Biology, Key Laboratory of Veterinary Parasitology of Gansu Province, Lanzhou Veterinary Research Institute, Chinese Academy of Agricultural Sciences, Lanzhou 730046, China; zouyangdr@163.com; 3College of Veterinary Medicine, Shanxi Agricultural University, Taigu 030801, China; xingquanzhu1@hotmail.com; 4Key Laboratory of Veterinary Public Health of Yunnan Province, College of Veterinary Medicine, Yunnan Agricultural University, Kunming 650201, China

**Keywords:** *Blastocystis* sp., prevalence, subtype, diarrheic pig, southern China

## Abstract

*Blastocystis* sp. is a common pathogen that infects the intestines of humans and animals, causing a threat to public health. However, little information on the prevalence and subtypes of *Blastocystis* sp. in diarrheic pigs in China is available. Herein, 1254 fecal samples were collected from diarrheic pigs in 37 intensive pig farms in Hunan, Jiangxi, and Fujian provinces in southern China, and the prevalence and subtypes of *Blastocystis* sp. were investigated. *Blastocystis* sp. was detected by PCR assay, which amplified the small subunit rRNA (SSU rRNA) gene. Overall prevalence of *Blastocystis* sp. was 31.4% (394/1254), including 21.5% (66/307), 33.1% (99/299), 58.9% (56/95), and 31.3% (173/553) in suckling piglets, weaned piglets, fattening pigs, and sows, respectively. Moreover, age and region factors were significantly related to prevalence of *Blastocystis* sp. (*p* < 0.05). Four *Blastocystis* sp. subtypes were identified, including ST1, ST3, ST5, and ST14. The preponderant subtype was ST5 (76.9%, 303/394). To our knowledge, ST14 was firstly found in pigs in China. The human-pathogenic subtypes (ST1, ST3, ST5, and ST14) that were observed in this study indicate a potential threat to public health. These findings provided a new sight for studying the genetic structure of *Blastocystis* sp.

## 1. Introduction

*Blastocystis* sp. is a zoonotic intestinal protozoan with a worldwide distribution. The host range of *Blastocystis* sp. is extensive, including humans, non-human primates, mammals, birds, fish, annelids, arthropods, reptiles, and amphibians [1]. Since the term “*Blastocystis*” was introduced by A. Alexieff in 1911, there has been a consensus that *Blastocystis* sp. is transmitted through the oral–fecal route, although its pathogenicity has been controversial [2,3]. *Blastocystis* sp. infection is in some cases thought to be associated with clinical symptoms, including abdominal pain, diarrhea, nausea, irritable bowel syndrome (IBS), and inflammatory bowel disease (IBD), which cause significant physical discomfort to human and animals [4,5,6,7]. Furthermore, in an infected host, *Blastocystis* sp. infection may concur with other zoonotic parasites such as *Giardia duodenalis* and *Cryptosporidium* spp. [8,9,10]. Hence, investigation of the prevalence and subtypes of *Blastocystis* sp. plays an important role in tracking and preventing the transmission of this protist.

Currently, based on molecular analysis of the small subunit ribosomal RNA (SSU rRNA), 27 subtypes of *Blastocystis* sp. have been identified, 12 of which (ST1–ST10, ST12, and ST14) were found in humans and animals, while the others were present in specific animals [9,10,11,12,13]. The prevalence of *Blastocystis* sp. varied considerably among different animals [14,15,16,17,18]. Furthermore, the dominant subtype of *Blastocystis* sp. in pigs is ST5, and ST10 is the dominant *Blastocystis* subtype in alpacas, while ST3 is the dominant *Blastocystis* subtype in humans [5,9,19]. ST14 appears more frequently in cattle and sheep [20]. The prevalence of *Blastocystis* sp. in pigs is varied in different regions [14,21].

Previous studies have reported high prevalence of *Blastocystis* sp. in humans, domestic animals, or wild animals in several provinces of China [22]. Prevalence of *Blastocystis* sp. in pigs has been reported in Shaanxi, Guangdong, Zhejiang, Heilongjiang, Jiangxi, and Yunnan provinces and Xinjiang Hui Autonomous Region in China [6,7,9,17,23,24]. However, there is no report of *Blastocystis* sp. infection in pigs in Hunan and Fujian provinces in China. Although there was a previous report of pig infection with *Blastocystis* sp. in Jiangxi Province [23], the sample size was too small, and might not have reflected the true situation of pigs infected with *Blastocystis* sp. Therefore, this study examined the prevalence of *Blastocystis* sp. and its subtypes in diarrheic pigs of different age groups and regions in three southern provinces of China.

## 2. Results

### 2.1. Prevalence of Blastocystis sp. in Diarrheic Pigs

In the present study, 31.4% (394/1254, 95% CI 28.85–33.99) of the 1254 fecal samples of diarrheic pigs were positive for *Blastocystis* sp. The prevalence of *Blastocystis* sp. in pigs in Fujian Province (43.7%, 59/135, 95% CI 35.34–52.07) and Jiangxi Province (30.5%, 316/1036, 95% CI 27.70–33.31) was higher than in Hunan Province (22.9%, 19/83, 95% CI 13.85–31.93), with ORs of 2.62 (95% CI 1.41–4.84) and 1.47 (95% CI 0.86–2.49), respectively (Table 1). In Fujian Province, the highest prevalence of *Blastocystis* sp. was found in pigs in Sanming City, with a positive rate of 45.2% (56/124, 95% CI 36.40–53.92). In Hunan Province, the highest *Blastocystis* prevalence was found in Zhuzhou City, with a positive rate of 41.7% (10/24, 95% CI 21.94–61.39). In Jiangxi Province, the highest *Blastocystis* prevalence was found in Fuzhou, with a positive rate of 47.2% (42/89, 95% CI 36.82–57.56) (Table 2). Significant differences (*p* < 0.05) in the prevalence of *Blastocystis* sp. in pigs in the three investigated provinces and in different cities of Jiangxi province were observed (Table 1 and Table 2).

Among different growing stages of pigs, fattening pigs had the highest prevalence (58.9%, 56/95, 95% CI 49.06–68.84) of *Blastocystis* sp., followed by weaned piglets (33.1%, 99/299, 95% CI 27.78–38.44), sows (31.3%, 173/553, 95% CI 27.42–35.15), and suckling piglets (21.5%, 66/307, 95% CI 16.90–26.09), and the differences were statistically significant (*p* < 0.001) (Table 1). Furthermore, fattening pigs had 5.24 times (95% CI 3.21–8.57) more risk of infection with *Blastocystis* than that of suckling piglets.

### 2.2. Subtype Distribution of Blastocystis sp. in Diarrheic Pigs

A total of four subtypes (ST1, ST3, ST5, and ST14) of *Blastocystis* sp. were identified in all growing stages of pigs in this study. Of these subtypes, ST5 was the predominant subtype of *Blastocystis* sp., with an infection rate of 76.9% (303/394), followed by ST3 (11.9%, 47/394), ST1 (6.6%, 26/394), and ST14 (4.6%, 18/394). Moreover, the ST14 was first identified in pigs in Fujian (3/18) and Jiangxi Provinces (15/18) (Table 1).

### 2.3. Phylogenetic Analysis of Blastocystis sp.

Phylogenetic analyses showed that the sequences of the four subtypes (ST1, ST3, ST5, and ST14) obtained from pigs in this study clustered with other ST1, ST3, ST5, and ST14 sequences obtained from other animals or humans into one branch, with high bootstrap values (Figure 1). Notably, The ST1 sequences (MW767060–MW767062) obtained from pigs in this study were closely related to the ST1 sequence (MK719635) obtained from humans (Figure 1). Furthermore, the ST14 sequences obtained from pigs in this study showed a closer genetic relationship with other ST14 sequences from ruminants (Figure 2).

## 3. Discussion

Although *Blastocystis* sp. has been researched for more than a century, its pathogenicity remains controversial [6,7]. There is not enough evidence for the clinical importance of *Blastocystis* sp., but its potential pathogenicity has long been studied [9]. Therefore, extensive investigation of *Blastocystis* sp. may improve the understanding of its pathogenicity and lead to effective prevention and control.

The overall prevalence of *Blastocystis* sp. in diarrheic pigs in Hunan, Jiangxi, and Fujian provinces was 31.4% (394/1254, 95% CI 28.85–33.99). Compared with the prevalence of *Blastocystis* sp. in healthy pigs in other studies, it was lower than that in Zimbabwe (82.8%) [25], Thailand (76.9%) [26], the Philippines (100%) [27], the USA (100%) [28], Australia (76.7%) [5], Cambodia (45.2%) [29], Indonesia (87.1%) [30], Brazil (44.4%) [31], Korea (59.3%) [32], the Czech Republic (89.8%) [25], and Germany (60.0%) [33]. It also was lower than that in Jiangxi Province (100%) [23], Shaanxi Province (74.8%) [9], Guangdong Province (55.6%) [7], and Yunnan Province (50.0%) [7,24] in China, but higher than that in Spain (7.5%) [14], the UK (28.5%) [34], Xinjiang Hui Autonomous Region (21.7%) [6], and Heilongjiang Province (8.80%) [35]. However, we did not collect fecal samples from healthy pigs in this study, making it impossible to directly compare the prevalence of *Blastocystis* sp. in healthy and diarrheic pigs. It is possible that no difference of prevalence could be found in health and diarrheic pigs. This finding reveals that diarrhea may not be associated with prevalence of *Blastocystis* sp., but more evidence is needed explain this finding. In addition, many factors may also contribute to the varying prevalence of *Blastocystis* sp., such as different detection methods, breeds, geography, or sample sizes. However, these pigs’ samples were previously tested for viruses and other pathogens, including PEDV, PDCoV, and *Escherichia coli* [36,37]. Several pathogens, especially helminths and protozoa, which are included among the most frequent causes of diarrhea in pigs, were not investigated, which was a limitation of the current study.

In previous reports, the prevalence of *Blastocystis* sp. in sows was generally significantly higher than that in fattening pigs, but in this study, the *Blastocystis* prevalence in fattening pigs was higher than that in other growing stages of pigs, which was consistent with the results of other studies [7,9,32]. This difference may have been caused by the raising condition. Furthermore, the infection rate of *Blastocystis* sp. was the lowest in suckling piglets compared to other growing stages of pigs, which might have been related to the important role of maternal antibodies.

Comparing the infection rate of ST3 with previous reports [5,9,15,21,32], we found that ST3 had the highest infection rate in this study (Table 1). According to previous studies, ST5 was widely distributed in all age groups of pigs, and only ST5 was detected in sows [6,9]. However, we found ST1, ST3, ST5, and ST14 in all age groups. Diarrhea often destroys the structure of the intestinal flora and alters the intestinal environment, which might lead to the change in dominant subtype [32]. This might explain the higher positive rate of ST3 compared to ST1 in diarrheic pigs in this study. Although the available data for ST14 in pigs is very limited [20], ST14 has been reported in humans [11], suggesting that ST14 has zoonotic potential, and that pigs may be a link in the transmission of ST14. Mixed infections were not detected in this investigation. While primers for subtype-specific detection are now available, only a limited number of subtypes are currently available for detection [17].

Phylogenetic analysis revealed that sequences of ST1 obtained from pigs in this study were closely related to human-derived sequences of ST1 (MK719635) (Figure 1), which further proved that pigs could be a possible reservoir for human infection with *Blastocystis* sp. The sequences of ST14 obtained in pigs and sequences of ST14 isolated in ruminants were clustered together (Figure 2). Since ST14 is more prevalent in ruminants, we speculated that the source of ST14 infection in pigs might be from ruminants. These findings should be enhanced in future molecular epidemiological studies.

## 4. Materials and Methods

### 4.1. Study Sites

The fecal samples of diarrheic pigs were collected from 37 intensive pig farms located in three southern provinces of China: Hunan (location: 24°38′–30°08′ N, 108°47′–114°15′ E), Jiangxi (location: 24°29′–30°04′ N, 113°34′–118°28′ E), and Fujian (location: 23°30′–28°22′ N, 115°50′–120°40′ E).

### 4.2. Sampling

From 2015 to 2019, a total of 1254 fresh fecal samples from pigs with diarrhea were collected from Jiangxi Province (n = 1036), Hunan Province (n = 83), and Fujian Province (n = 135) (Table 1 and Figure 3). Only pigs with dilute feces or watery diarrhea were sampled, and all samples were collected by anal swab. Among these fecal samples, 307 fecal samples were from suckling piglets (<21 days), 299 fecal samples were from weaned piglets (21–70 days), 95 fecal samples were from fattening pigs (71–180 days), and 553 fecal samples were from sows (>180 days) (Table 1). All fecal samples were placed in a cryopreservation box with an adequate amount of ice bags immediately after sampling, and then were stored in a refrigerator at −80 ℃ before DNA extraction.

### 4.3. Genomic DNA Extraction and PCR Amplification

Approximately 300 mg of each fecal sample was washed 3 times with distilled water by centrifuging at 13,000× *g* for 5 min to remove the impurities. The remaining sediments were used to extract the genomic DNA using the E.Z.N.A.^®^ Fecal DNA Kit (D4015-02, Omega Bio-Tek Inc., Norcross, GA, USA). The genomic DNA was stored at −20 ℃ for further analysis. The genomic DNA samples were screened for *Blastocystis* sp. by PCR amplification of the SSU rRNA genes with a target fragment size of roughly 600 using the primers BhRDr (5′-GAGCTTTTTAACTGCAACAACG-3′) and RD5 (5′-ATCTGGTTGATCCTGCCAGT-3′) [9]. The 25 μL reaction system contained 2 μL of genomic DNA, 0.2 mM of dNTP mixture, 1.5 mM of MgCl2, 2.5 μL of 10× Ex Taq buffer, 1.25 U of TaKaRa Ex Taq® (Takara Bio Inc., Dalian, China), and 0.25 μL of primers (10 mol/μL). The PCR reaction conditions were set as follows: initial denaturation at 94 °C for 5 min; 35 cycles at 94 °C for 45 s, 59 °C for 45 s, and 72 °C for 1 min; and an additional 72 °C extension for 3 min. Each reaction included a positive (DNA from *Blastocystis* sp.) and a negative control (reagent water). The final PCR products were identified by 2% (*w*/*v*) agarose gel electrophoresis and stained with ethidium bromide.

### 4.4. Sequence Analysis

Approximately 600 bp PCR product of each sample was recovered and purified by Tsingke Biotechnology Co., Ltd. for sequencing using the Sanger sequencing method. The subtypes of *Blastocystis* sp. were identified by aligning with obtained sequences and corresponding sequences available in the GenBank database (http://www.ncbi.lm.nih.gov/GenBank/, accessed on 21 March 2021) using Clustal X 2.1 (http://www.clustal.org/clustal2/, accessed on 28 March 2021) [26]. The maximum likelihood (ML) method with a Kimura 2-parameter model in MEGA 7.0 (http://www.megasoftware.net accessed on 1 September 2021) [27] was used to establish a phylogenetic tree with 1000 repeats under a bootstrap method, and U21338 was set as the out-group (Figure 1 and Figure 2).

### 4.5. Statistical Analysis

Data obtained in this study on the prevalence of *Blastocystis* sp. between different regions and age groups were systematically analyzed with a chi-square test (χ^2^) using SPSS version 25.0 (IBM SPSS Inc., Chicago, IL, USA), and significant differences were considered only when the obtained *p* value was less than 0.05.

## 5. Conclusions

In the present study, a total of 1254 fecal samples from diarrheic pigs in three provinces in southern China were examined for the prevalence and subtypes of *Blastocystis* sp. This was the first report on *Blastocystis* sp. infection in pigs in Hunan and Fujian provinces. Three zoonotic subtypes (ST1, ST3, and ST5) and one potential zoonotic subtype (ST14) were identified, and ST14 was detected for the first time in pigs in China. Compared to previous reports of healthy pigs infected with *Blastocystis* sp., the great differences found in the present study were reflected in the increased frequency of ST1 and ST3, the significantly higher *Blastocystis* sp. infection in fattening pigs rather than in sows, and the detection of ST14. These findings may help understand the genetic structure of *Blastocystis* sp. in pigs, providing useful data for effective prevention and control of *Blastocystis* sp. in southern China in the future.

## Figures and Tables

**Figure 1 pathogens-10-01189-f001:**
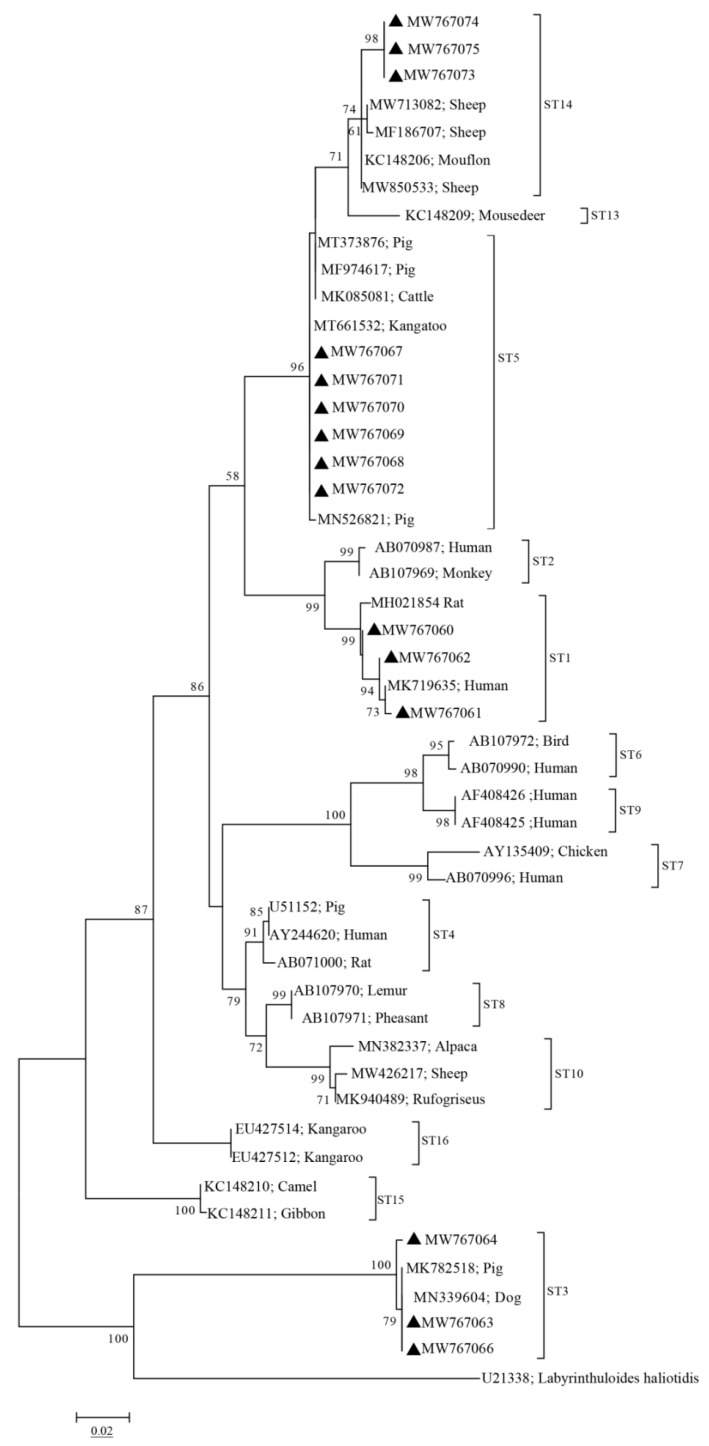
Phylogenetic analysis of *Blastocystis* subtypes based on sequences of the SSU rRNA gene using the maximum likelihood (ML) method. A bootstrap algorithm was used to assess the branch reliability with 1000 replicates. Only bootstrap values above 50% are shown. Sequences marked with black triangles indicate the sequences obtained in this study, and the GenBank accession numbers of the sequences are shown to the right of the triangle.

**Figure 2 pathogens-10-01189-f002:**
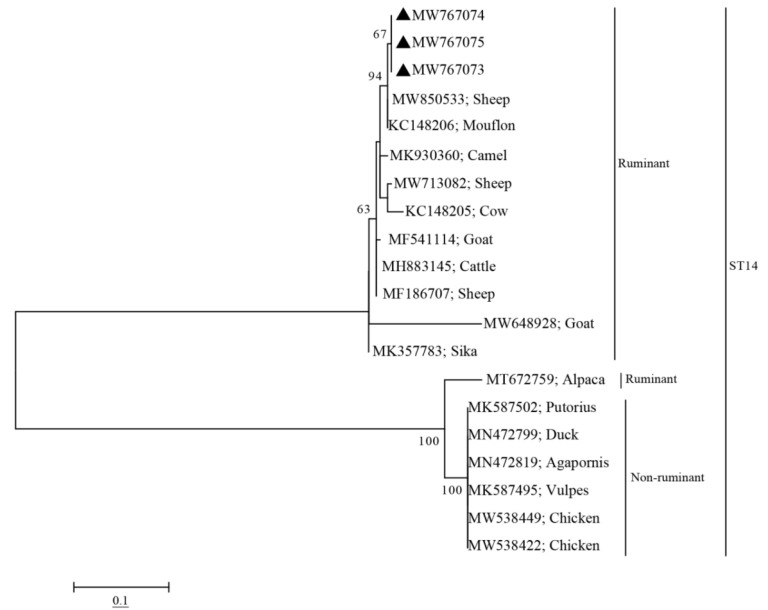
Phylogenetic analysis of *Blastocystis* ST14 based on the SSU rRNA gene. The model and parameter settings for constructing the phylogenetic evolutionary tree were same as those used for constructing Figure 1. Sequences marked with black triangles indicate the sequences isolated in this study, and the GenBank accession numbers of the sequences are shown to the right of the triangle.

**Figure 3 pathogens-10-01189-f003:**
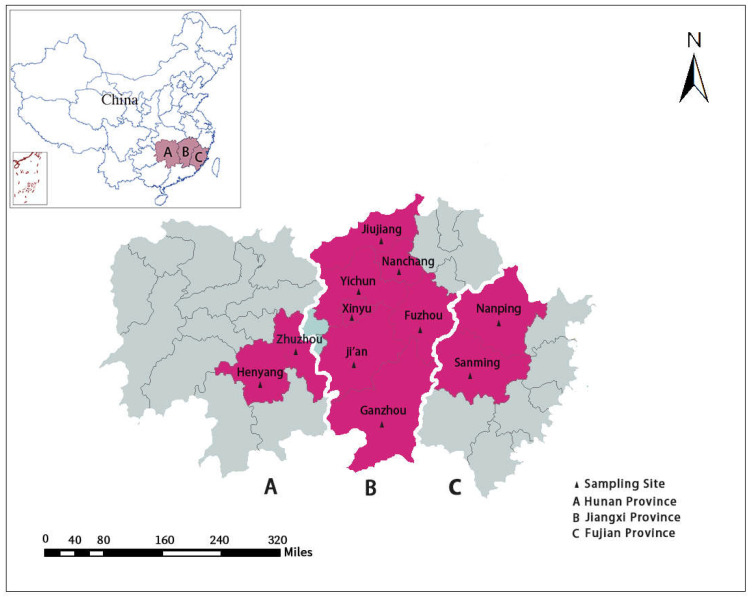
The map of the sample collection in this study; the red areas represent the sampling locations.

**Table 1 pathogens-10-01189-t001:** Factors associated with prevalence and subtype distribution of *Blastocystis* sp. in pigs with diarrhea in southern China.

Factor	Category	No. Tested	No. Positive (%) (95% CI)	OR (95% CI)	*p* Value	Subtype (No.)
Region	Hunan	83	19 (22.89) (13.85–31.93)	Reference	0.002	ST1 (1), ST3 (2), ST5 (16)
	Jiangxi	1036	316 (30.50) (27.70–33.31)	1.47 (0.86–2.49)		ST1 (23), ST3 (36), ST5 (242), ST14 (15)
	Fujian	135	59 (43.70) (35.34–52.07)	2.62 (1.41–4.84)		ST1 (2), ST3 (9), ST5 (45), ST14 (3)
Age	Suckling piglets (<21 days)	307	66 (21.50) (16.90–26.09)	Reference	<0.001	ST1 (9), ST3 (13), ST5 (40), ST14 (4)
	Weaned piglets (21–70 days)	299	99 (33.11) (27.78–38.44)	1.81 (1.26–2.60)		ST1 (5), ST3 (9), ST5 (82), S14 (3)
	Fattening pigs (71–180 days)	95	56 (58.95) (49.06–68.84)	5.24 (3.21–8.57)		ST1 (4), ST3(5), ST5 (42), ST14 (5)
	Sows (>180 days)	553	173 (31.28) (27.42–35.15)	1.66 (1.20–2.30)		ST1 (8), ST3 (20), ST5 (139), ST14 (6)
Total		1254	394 (31.42) (28.85-33.99)			ST1 (26), ST3 (47), ST5 (303), ST14 (18)

**Table 2 pathogens-10-01189-t002:** Distribution of *Blastocystis* sp. subtype in different sampling cities.

Province	City	No. Tested	No. Positive (%) (95% CI)	OR (95% CI)	*p* Value	Subtype (No.)
Hunan	Hengyang	59	9 (15.3) (6.08–24.43)	Reference	0.009	ST3(2), ST5(7)
	Zhuzhou	24	10 (41.7) (21.94–61.39)	3.97 (1.35–11.66)		ST1(1), ST5(9)
Jiangxi	Yichun	236	46 (19.5) (14.44–24.55)	Reference	<0.001	ST1(2), ST3(7), ST5(34), ST14(3)
	Ganzhou	38	10 (26.3) (12.32–40.32)	1.48 (0.67–3.25)		ST1(1), ST3(4), ST5(5)
	Nanchang	173	48 (27.7) (21.07–34.42)	1.59 (1.00–2.52)		ST1(3), ST3(10), ST5(35)
	Ji’an	185	53 (28.6) (22.13–35.16)	1.66 (1.05–2.61)		ST1(7), ST3(7), ST5(36), ST14(3)
	Xinyu	76	23 (30.3) (19.93–40.59)	1.79 (1.00–3.22)		ST1(3), ST5(20)
	Jiujiang	239	94 (39.3) (33.14–45.52)	2.66 (1.77–4.05)		ST1(5), ST3(4), ST5(80), ST14(5)
	Fuzhou	89	42 (47.2) (36.82–57.56)	3.69 (2.18–6.25)		ST1(2), ST3(4), ST5(32), ST14(4)
Fujian	Nanping	11	3 (27.3) (0.95–53.95)	Reference	0.250	ST3(2), ST5(1)
	Sanming	124	56 (45.2) (36.40–53.92)	2.20 (0.56–8.67)		ST1(2), ST3(7), ST5(44), ST14(3)
Total		1254	394 (31.42) (28.85–33.99)			ST1 (26), ST3 (47), ST5 (303), ST14 (18)

## Data Availability

For reasonable requests, the data obtained for this study can be obtained by contacting the corresponding author. The sequences of the *Blastocystis* sp. obtained from this study were deposited in the NCBI GenBank database under the accession numbers MW767060–MW767075.

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
