# Peer review of "Prevalence and Subtype Distribution of Blastocystis sp. in Diarrheic Pigs in Southern China"

_pathogens, 2021, doi:10.3390/pathogens10091189_

Round 1

Reviewer 1 Report

The issue is current. Blastocystosis has not yet been sufficiently studied. Based on the breeding of pigs for consumption and the associated transport around the world, monitoring of these diseases is also important. Especially if they endanger human health due to the potential zoonotic risk. It is important to continue in the future in the identification of individual subtypes, due to the possible explanation of the occurrence of individual STs in certain groups of animals. This may indicate specificity and affinity for a particular group.

Comments:

Abstract: OK

Introduction: OK

Results and Tables: OK

Discusion: OK

Materials and Methods

line 193: Volume of water used..

               And why only water??

Discusion:

line 133-141: Complete citations. Not just Asia and 1 state in Europe.
                      It's important because of the export of pigs

line 164: not ..."ST 14 has zoonotic potential.." to change ...." zoonotic                                  transmission is possible..."  

Conclusion: OK

Author Response

Responses to comments and suggestions of Reviewer #1:

General comments:

The issue is current. Blastocystosis has not yet been sufficiently studied. Based on the breeding of pigs for consumption and the associated transport around the world, monitoring of these diseases is also important. Especially if they endanger human health due to the potential zoonotic risk. It is important to continue in the future in the identification of individual subtypes, due to the possible explanation of the occurrence of individual STs in certain groups of animals. This may indicate specificity and affinity for a particular group.

Response: We thank Reviewer #1 very much for favorable comments and suggestions on MS. 

Minor point:

Q1:Materials and Methods. line 193: Volume of water used.And why only water??

Response: We thank Reviewer #1 very much for the query. We have added the distilled water up to 2 ml in each fecal sample. The fecal samples have many water-soluble impurities, therefore, only distilled water was used to remove it. 

Q2: line 133-141: Complete citations. Not just Asia and 1 state in Europe. It’s important because of the export of pigs."

Response: We thank Reviewer #1 very much for the query. We have added other studies on the prevalence of Blastocystis sp. in pigs in the UK, Czech and Germany in the section of discussion. 

Q3: Line 164: not..."ST14 has zoonotic potential" to change..."zoonotic transmission is possible...".

Response: We thank Reviewer #1 very much for the suggestion. We have changed this sentence accordingly. 

We sincerely hope that you find our MS revised to your satisfaction. We are looking forward to receiving your editorial decision soon and hope to see our work published in Pathogens.

With best wishes,

Xiao-Qing Chen,

On behalf of all co-authors.

Reviewer 2 Report

The manuscript “Prevalence and Subtype Distribution of Blastocystis sp. in Diarrheic Pigs in Southern China” reports on the occurrence and subtype diversity of Blastocysits from 37 intensive pig farms in Hunan, Jiangxi and Fujian provinces in southern China. A large number of samples were recovered. Overall, the results are of interest and value to the Blastocystis community as they contribute to our understanding of Blastocystis subtype distribution. I suggest this paper be accepted for publication following some minor revision. However, the main doubt/perplexity is about the use of NJ method. I suggest to perform the analysis with ML and/or BI that are the two most robust. My specific minor comments are:

  • Line 19: delete “in” before the intestines
  • Line 47: please modify “the pathogenicity of this pathogen” in “its pathogenicity”
  • Line 60: please add citations: Gabrielli et al., 2021 Microorganisms
  • Table 2. enlarge the table
  • Line 99: change “has” with “had”
  • Line 107: add “of”
  • Line 109: delete “were”
  • Line 119: change “will be” with “are”
  • Line 126: remove the capitalization to “were”

In M&M you could add a map with sampling location and samples number per area

  • Line 200: add citation to primers
  • Line 209: add purification information
  • Line 280: modify “eiversity”

Author Response

Responses to comments and suggestions of Reviewer #2:

General comments:

The manuscript “Prevalence and Subtype Distribution of Blastocystis sp. in Diarrheic Pigs in Southern China” reports on the occurrence and subtype diversity of Blastocysits from 37 intensive pig farms in Hunan, Jiangxi and Fujian provinces in southern China. A large number of samples were recovered. Overall, the results are of interest and value to the Blastocystis community as they contribute to our understanding of Blastocystis subtype distribution. I suggest this paper be accepted for publication following some minor revision. However, the main doubt/perplexity is about the use of NJ method. I suggest to perform the analysis with ML and/or BI that are the two most robust. My specific minor comments are:

Response: We thank Reviewer #2 very much for favorable comments and suggestions on MS. We have used the ML method to reconstruct the phylogenetic tree.

Minor point:

Q1:Line 19: delete “in” before the intestines

Response: Added accordingly.

Q2: Line 47: please modify “the pathogenicity of this pathogen” in “its pathogenicity”

Response: We thank Reviewer #2 very much for the suggestions. We have revised this sentence.

Q3: Line 60: please add citations: Gabrielli et al., 2021 Microorganisms

Response: Added accordingly.

Q4: Table 2. enlarge the table

Response: We thank Reviewer #2 very much for the suggestions. We have enlarged the table 2.

Q5: Line 99: change “has” with “had”

Response: Changed accordingly.

Q6: Line 107: add “of”

Response: Added accordingly.

Q7: Line 109: delete “were”

Response: Deleted accordingly.

Q8:Line 119: change “will be” with “are”

Response: Changed accordingly.

Q9:Line 119: change “will be” with “are”

Response: Changed accordingly.

Q10:Line 126: remove the capitalization to “were”

Response: Removed accordingly.

Q11:In M&M you could add a map with sampling location and samples number per area

Response:We thank Reviewer #2 very much for the suggestions. We have added the map with sampling location and samples number per area. See Figure 3.

Q12:Line 200: add citation to primers

Response: Added accordingly.

Q13:Line 209: add purification information

Response: Added accordingly.

Q14:Line 280: modify “eiversity”

Response: Changed accordingly.

We sincerely hope that you find our MS revised to your satisfaction. We are looking forward to receiving your editorial decision soon and hope to see our work published in Pathogens.

With best wishes,

Xiao-Qing Chen,

On behalf of all co-authors.